# Building Adaptation Measures Using Future Climate Scenarios—A Scoping Review of Uncertainty Treatment and Communication

Jørn Emil Gaarder [1,*], Hans Olav Hygen [2], Rolf André Bohne [1] and Tore Kvande [1]

1  Department of Civil and Environmental Engineering, Norwegian University of Technology and Science (NTNU), 7030 Trondheim, Norway
2  Norwegian Meteorological Institute (MET), 0313 Oslo, Norway
*  Correspondence: jorn.e.gaarder@ntnu.no

**Abstract:** The global climate is changing. Predicting the impacts this will have on buildings is the first step in the process of finding suitable building adaptation measures. Future climate adaptation of buildings and infrastructure is a growing field of research, relying on both socio-economical and meteorological research for input values to the simulation models. Models producing hourly future weather data rely on global climate models which are based on emission scenarios made from assumptions of future political, social, and economic developments. Accounting for the uncertainties from these underlying models as much as possible, and communicating the uncertainties in the results, is obviously paramount for reliable conclusions from the building simulation models. This paper is a scoping review, investigating how 132 studies treat and communicate the string of uncertainties from underlying models connected to future weather file generation in the scientific literature on building adaptation research. The findings suggest that climate-model-induced uncertainties are often under-communicated, due to either insufficient analysis or neglect. The studies that included the most comprehensive analyses of the uncertainties frequently concluded that treatment of these is important for the reliability of the results, and neglecting this could lead to misleading conclusions.

**Keywords:** uncertainty; emission scenario; GCM; climate model; building adaptation measure

## 1. Introduction

### 1.1. Background

Climate change is one of the main challenges we are facing in the 21st century, with wide-spanning environmental, economic, and health related consequences [1]. The built environment is a considerable contributor to greenhouse gas emissions by being responsible for 25–40% of world-wide emissions [2]. It is also a victim of the consequences, as climate effects are expected to lead to costly damages to buildings and infrastructure in the years to come [3]. Reliable tools for assessing building performance under future climatic strains are needed if we are to adapt to the future climate [4]. Research into building adaptation measures using future climate scenarios is paramount in such a tool kit, requiring input contributions from socio-economical and meteorological research for production of emission scenarios and climate models. However, these underlying models will introduce a higher level of uncertainty to the results when compared to using the present-day climate as the model input. Assessment of how future climate-induced uncertainties may influence the results in such studies places extra demands on the comprehensiveness of the method. Further, implementing uncertainty evaluations into the presentation of results complicates conclusions, distracts from the main findings, and are challenging to communicate in a clear way. For these reasons, implementation of uncertainty evaluations in the methodology may be neglected in many cases. Uncertainty is a wide-ranging concept which comes in many forms, and methods for managing it will be decided by the properties of the

particular uncertainty. Perceptions of risk and uncertainty are subjective in nature, and as such they will be formed by the attributes of the perceiver, and the characteristics of the risk. There are two fundamental pathways in which human beings comprehend risk and uncertainty, namely the analytic system and the experiential system [5]. The analytic system is characterized by using algorithms and normative rules, while the experiential system relies on associations, emotions, and affect. The analytic system for uncertainty treatment requires the creation of a model of the problem at hand, and Riesch et al. have proposed to categorize uncertainties related to such a model into five distinct levels, as outlined in Table 1 [6]. Note that the term "model" in Table 1 refers to any model of empirical reality, and not climate models or emission scenario models specifically.

**Table 1.** Hierarchical levels of uncertainty in prospects [6].

| Category | Description |
|---|---|
| Level 1 | Uncertainty about the outcome: The model is known, the parameters are known, and it predicts a certain outcome with a probability $p$. |
| Level 2 | Uncertainty about the parameters: The model is known, but its parameters are not known. A lack of empirical information leads to uncertainties in the probability of the outcome. |
| Level 3 | Uncertainty about the model choice: There are several models to choose from, and we have an idea of how likely each competing model reflects reality. |
| Level 4 | Uncertainty about acknowledged inadequacies and our implicitly-made assumptions in the chosen model: The validity of the assumptions is questioned, both omitted and included in the model. |
| Level 5 | Uncertainty about unknown inadequacies: There is uncertainty as to whether a high impact parameter in the model is lacking because the parameter is unknown. |

Uncertainties in emission scenarios and future climate modelling are dominated by levels 4 and 5. There are several models to choose from, but the model that most accurately represents reality cannot be verified because of the vast complexity of the parameters and assumptions [6]. Some of the uncertainties in the global climate simulation models are quantifiable and manageable by means of empirical and computational improvements, while others represent unquantifiable, irreducible epistemological limits related to inductive reasoning and to the nature of model-based global science [7]. To mitigate this problem, a parameter of uncertainty can be introduced that may provide the ground for practical decision making by incorporating the uncertainties in such a way that quantitative estimates may be given [8]. Ignoring this uncertainty when using future climate models, on the other hand, will carry the higher-level uncertainties of Table 1 from the climate models into the results.

This study seeks to map the extent to which climate-scenario-induced uncertainties are evaluated and communicated in building climate adaptation research through a scoping review of the published literature in the field of climate adaptation measures for buildings from 2014 to 2021. Building climate adaptation is defined in this study as measures taken to prepare buildings for future changes in climatic strains, including their effects on indoor climate and energy use. The selection of studies is based on the findings from a previous scoping review performed by Stagrum [9,10], where the extent of scientific publications on climate adaptation within the field of building physics was mapped. The literature found by Stagrum is updated here with the published research from 2019–2021, using the same search method. The selection of studies is restricted by only considering the literature that employ the use of future climate scenarios for adaptation measures of a building and its parts against climatic strains, including future climate effects on indoor climate and building energy use.

The purpose of this work is to first map the methods used for creating future climate scenarios in building adaptation research, and then review how sources of uncertainties related to future climate estimation are principally treated and communicated. The focus of the study is the awareness, and ultimate communication, of the uncertainties generated through future climate modelling. The following research questions are addressed:

1.  Which methods are used for producing future climate scenarios in research concerning future climate adaptation measures for buildings?
2.  How are climate-model-induced uncertainties influencing the results evaluated and communicated?

This study is directed towards scientists within the field of building climate adaptations, aiming to increase awareness of the underlying uncertainties involved in future climate projections. Treatment and communication of these uncertainties are important and, as the results will show, often neglected or underestimated. Specific methods for numerically treating and mitigating uncertainties in future climate models is outside the scope of this review, as the focus is to see the extent to which they are principally treated through methodological choices and ultimately communicated through the results. Fortunately, there are several excellent works addressing the technical challenges in mitigating uncertainties in future weather file synthesis, some of which are presented in Sections 1.2–1.4. The reader is encouraged to look through the referred studies in this section to find specific methods and tools for mitigating the uncertainty when using future climate models in building adaptation studies.

In the following sections, an overview of the field of future climate modelling is presented, with an emphasis on the uncertainties induced by the models, as a theoretical framework for the subsequent discussion of the review findings.

### 1.2. Emission Scenarios

In 2000, the Intergovernmental Panel on Climate Change (IPCC) published a set of future emission scenarios in their Special Report on Emission Scenarios (SRES) [11], in connection to their 4th assessment report (AR4). In 2014, these were revised in the IPCC's 5th assessment report (AR5) [12], with a set of Representative Concentration Pathways (RCP). The SRES scenarios from the 4th assessment report are named by family (A1, A2, B1, and B2), where each family is designed around a set of consistent assumptions, such as a world that is more integrated or more divided [13]. The RCP scenarios from the 5th assessment report, on the other hand, are named for the resulting change in radiative forcing in 2100, from +2.6 W/m$^2$ to +8.5 W/m$^2$. Comparing the $CO_2$ concentrations and the resulting global temperature change between the SRES and RCP scenarios, the SRES A1F1 is comparable to RCP 8.5; the SRES A1B to RCP 6.0; and the SRES B1 to RCP 4.5. The RCP 2.6 scenario is much lower than any SRES scenario because it includes policies to achieve net negative carbon dioxide emissions before the end of the century, while SRES scenarios do not [13]. The SSP scenarios used in CMIP6 and IPCC-AR6 are not relevant here since the cut-off for papers evaluated is prior to the launch of IPCC-AR6-WG1.

The emission scenarios have been used for the past decade by the scientific community for creating projections of future climate changes, as well as assessing the consequences for a given object of study. Numerous Global Circulation Model (GCM) projections based on these emission scenarios have been produced by different scientific communities across the globe [14–16].

### 1.3. Climate Models, Downscaling Methods, and Weather File Synthesis

The global models, as those used in the IPCC, give a good representation of possible futures at a global and regional level, but is far too coarse a tool on a national and sub national level; here, the various techniques for downscaling come into play. The core idea of downscaling climate projections is to combine the coarse regional/global results with more localized information, e.g., topography. There are in principle three approaches to this: (1) change factor method where one, e.g., scales the known climate at a location with the projected global change, (2) statistical downscaling, where the basic idea is to use statistical methods to connect global/regional weather patterns to a local response, and (3) dynamical downscaling, where one uses regional climate models to project the local climate and weather with border conditions from the global climate model [17]. All of these methods have advantages and disadvantages. The advantage of dynamic downscaling is

to be able to create datasets where extraction of, e.g., physically consistent time series is possible. The advantage of statistical downscaling, which produces a statistical distribution and needs a weather generator to create time series if that is required, is that speed and low cost facilitate a broader exploration of the full range of possible outcomes [18]. For a comprehensive review of the details in climate modelling methods Ekstrøm et al. [19] and Rummukainen [17] are recommended reads.

In order to translate future climate projections into a usable format for building climate adaptation studies, synthesis of weather files based on the climate models becomes necessary. A common method is to create a typical meteorological year, a method developed by Hall et al. in 1978 [20], where representative months from several years of observation or climate model output are selected and compiled. These are then broken down into daily and hourly values, relying on observed statistical distributions of daily indices such as minimum, mean, and maximum values of the climate variables. A major disadvantage to this method is that extreme conditions are underestimated due to the nature of averages. This disadvantage is even more prevalent when considering future climate changes, due to the increased occurrence and severity of extreme weather expected [21]. There are several propositions to mitigate this problem in the literature, e.g., using more than one synthesized weather file [22,23], where extreme conditions are taken into account. Comprehensive predeveloped data sets exist for many areas, such as in the UK where the UKP09 weather generator [24,25] provides a range of possible climate change projections based on 100 sets of 30-year-long synthetic weather data, thus providing probabilistic distributions for risk assessments [26].

### 1.4. Uncertainties in Climate Projections

Nils Bohr is commonly quoted with the words "Prediction is very difficult, especially if it's about the future!" and yet this is exactly what climate researchers try to do. The main tool for this is earth system models and atmosphere–ocean general circulation models, more commonly known as climate models. The aim of these models is to simulate the earth climate system as it has been and as it may be in the future. The earth system is complex and thus provides multiple sources of uncertainties. These uncertainties may originate in the simplification of the earth to the finite grid and the physics to fit into this grid; sub grid phenomena are usually implemented with statistics and parameterization. There is also a distinct natural variability in the models. Another set of uncertainties when projecting the future is which future to project. To compensate for the inherent uncertainties in the choice of scenarios, biases in global models, and downscaling techniques, the use of broad ensembles is common in climate research. A broader summary of the models' uncertainties may be found in the IPCC's "Climate Change 2021: The Physical Science Basis", especially chapter 4 by Lee et al. [27].

Ekstrøm et al. [19] classify most of the uncertainties surrounding global climate projection into three types: (1) Choice of emission scenario. (2) Uncertainties in the system's response, and how well the model simplifications replicate the earth's response. (3) Uncertainties in natural variability due to the chaotic nature of weather patterns. The different uncertainties introduced by emission scenario choices (type 1) and model selections (type 2) are largely connected to levels 3 and 4 in Riesch's categorization (see Table 1). Following the value chain of the climate projections, the GCM contains both errors and simplifications that may result in systematic biases. Choice of methods and domains for downscaling further add potential uncertainties and biases, which may be partly compensated for through bias corrections but will always include uncertainties due to natural variability (type 3). Besides bias corrections, careful selection of models may compensate for some of the shortcomings. The different uncertainties introduced in the methods used for creating future weather data will thus be connected to all levels in Riesch's categorization described in Table 1. However, the ones that may be the most easily treated by a building adaptation researcher are connected to levels 1–3, by model selection, model calibration, and model output treatment. Ideally, the sample of underlying models used should be as wide as possible to cover the

full range of possible futures under a chosen projection [28]. Figure 1 shows the effect of subsampling, where the blue and red represent the national climate projections for Norway, while the full range is presented in orange [28,29].

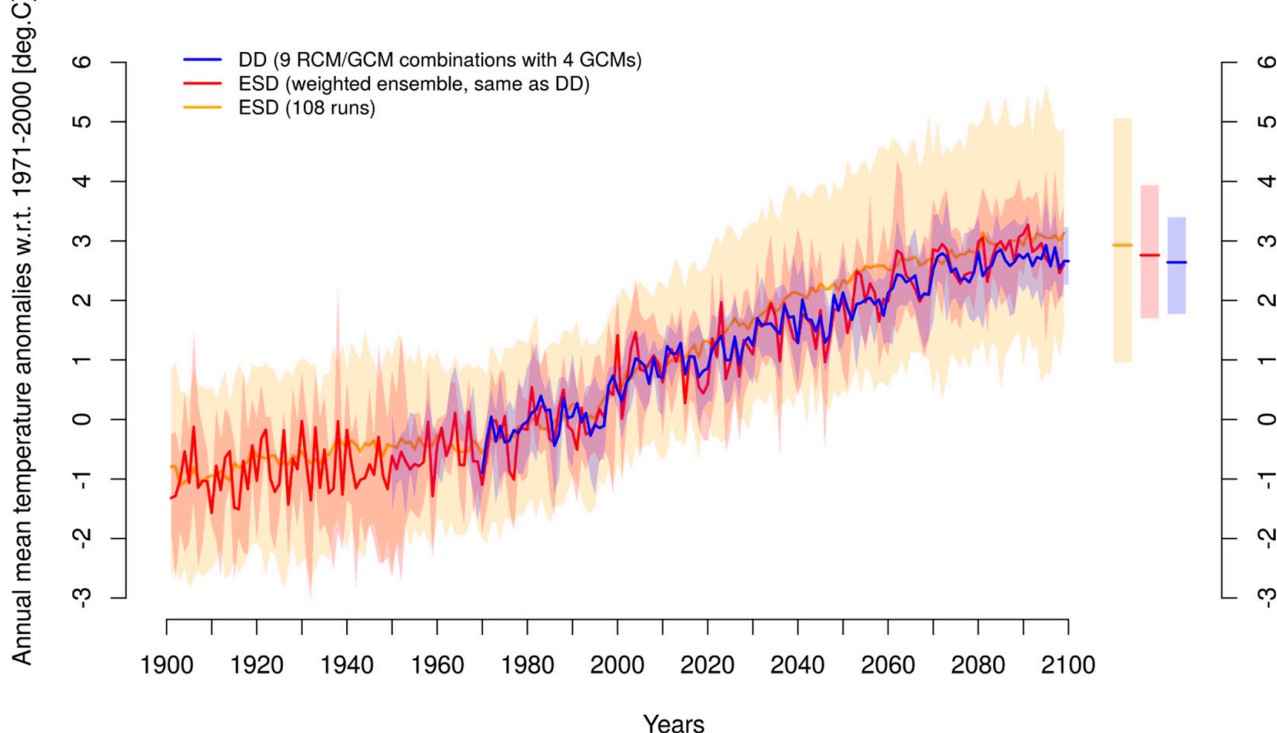

**Figure 1.** Historic and future climate projections for Norway under scenario RCP4.5, by empirical statistical downscaling. Blue represents nine projections for Norway that have been dynamically downscaled and bias-corrected, red is the same models statistically downscaled, and orange presents the full range of the GCMs statistically downscaled for Norway [28,29].

Considering RCP 4.5 and RCP 8.5, which are the two most frequently used scenarios found in this review, the difference in precipitation and temperature predictions has been shown to be larger between the scenarios than the variation between different GCMs for considerations of the 2071–2100 climate [30]. This means that emission-scenario-related uncertainties yield larger variation in final results than climate-model-related uncertainties.

All downscaling will carry the biases and errors of the respective GCM and add uncertainty from other sources as well. Model structure uncertainty is the main source of standard error in GCM simulation, and systematic bias correction by comparing model outputs to observations will reduce this error [31]. Wu et al. [32] demonstrate the inherent uncertainties from the driving conditions and the uncertainties in the RCM. Calculating an expression for the inherent and added uncertainty from dynamic downscaling is both difficult and valuable. Boundary conditions and initial conditions will both contribute to the uncertainty level of the model. However, as simulation lengths increase, the contribution from initial conditions will decrease and boundary condition contributions will dominate [32]. In an ideal world, increasing the resolution of the models would decrease the uncertainties in the output. However, increasing the resolution introduces new challenges, such as accurate prediction of local precipitation intensity over shifting terrain, and thus a careful evaluation of the required resolution for the problem at hand is necessary [32,33].

The GCM output of hydrological variables, such as precipitation levels and intensities, is wide-spread due to several uncertainties introduced at multiple stages [34]. Extreme rainfall can be particularly difficult to simulate on a local level because high resolution models of cloud formations and air movements are complex, with a high variability between the models, even on a global scale [32]. A comparison study of 20 GCMs under both RCP 4.5

and RCP 8.5 scenarios showed that the projection uncertainty of precipitation predictions reached 25.1% for RCP 4.5 and 33.5% for RCP 8.5 in 2071–2100 simulations. Overall, GCMs forced by high-emission scenarios yield higher variations between the models than GCMs forced by low-emission scenarios [30]. This emphasizes the importance of using multiple GCMs when producing weather data for climate adaptation studies, to produce a more complete description of possible futures that is not unnecessarily limited by subsampling.

The IPCC presents a common framework for presenting uncertainties in findings, to ensure consistent treatment and communication of uncertainties [35]. They suggest using a level of confidence scale based on a qualitative assessment of agreement and evidence for the findings. The nine possible confidence categories are combinations of low-medium-high agreement based on consistency and limited-medium-robust evidence based on type, amount, quality, and consistency. If the confidence level is sufficiently high, a confidence evaluation should be performed. If uncertainties can be quantified probabilistically, the findings can be further characterized by using either a consistent language for likelihood characterization suggested by the IPCC or probabilistic quantification of the uncertainties whenever possible [35]. As both emission scenarios and GCMs are very complex, a systematic approach to generalize uncertainty levels across different types of studies and findings proved to be difficult. A review of the papers using the IPCC guidance note highlighted three problems in the application of the procedure: (1) procedural unreliability stemming from nontransparent qualitative judgements with possible biases, (2) deep uncertainties related to epistemic approaches, broad scope of diverse problems and complex systems, and (3) communication of the uncertainties to users may be skewed, as information with a high degree of certainty is focused on, while information concerning large negative consequences is neglected if information uncertainty is high [36].

## 2. Materials and Methods

### 2.1. Overview

The scoping literature review presented in Sections 3 and 4 is conducted in accordance with the methodological framework described by Arksey and O'Malley [37], which is a six-step procedure: (1) identifying the research question, (2) identifying relevant studies, (3) study selection, (4) charting data, (5) collating, summarizing, and reporting the results, and (6) consultation.

### 2.2. Identifying and Selecting Relevant Studies

To ensure that the selection of literature from the last three years is compatible with the original literature gathered by Stagrum, the same method was employed. A summary of the search method used by Stagrum [9,10], modified to fit the added restrictions in scope in this study, is presented below.

Two search processes were used, hand searching of selected key journals and structured key word searching of selected databases. The selected journals are listed in Table 2, along with the selected key words and search engines used in the search. Searches were structured by combining two or more key words in order to reduce the number of hits to a manageable level (<200–300 approx.). Studies were first selected based on relevancy in the title, and then either included or excluded based on abstract relevancy. The relevant studies were limited to articles concerning adaptation measures in the field of building physics employing future climate scenarios from 2014 to 2021. Articles concerning urban and spatial planning, infrastructure, and governance were excluded. Articles where climate model method development were the main focus were also excluded, although some of the ultimately included articles were case studies with the aim of testing a newly developed methodology. These "grey-area" articles were included or excluded on a case-by-case basis, by determining whether the case study or the method development was the main focus of the article.

**Table 2.** Summary of journals, search engines, and key words used in the literature search.

| Journals | Search Engines | Key Words |
|---|---|---|
| Building and Environment<br>Climate Services<br>Energy and Buildings<br>Building Research & Information<br>Journal of Climate Change<br>Buildings<br>Journal of Building Physics<br>Sustainable Cities and Society<br>International Journal of Climate Change<br>Strategies and Management | Science Direct<br>Oria<br>Google Scholar | Climate change<br>Adaptation<br>Impact<br>Building<br>Energy<br>Thermal comfort<br>Cooling<br>Overheating<br>Measure<br>Retrofit |

Stagrum identified 82 studies using future climate scenarios in the original search from 2014–2018 [9,10]. Conducting the same search with the same limitations of scope for 2019–2021 yielded a further 50 studies, making the total sum of identified studies 132. The final selection of studies was processed in accordance with the method described by Arksey and O'Malley [37], with an emphasis on climate modelling methods and uncertainty assessments as per the research questions described in Section 1.1.

### 2.3. Charting Data

The methodology for producing the underlying climate models has been systematically analyzed for all studies by answering a set of key questions: (1) which emission scenarios are used, (2) what method is used for producing weather data, and (3) what climate parameters are needed to produce the models.

The treatment of climate-related uncertainties has been systematically analyzed in the same way by answering another set of key questions: (1) how is the emission scenario chosen and are uncertainties related to the choice evaluated, (2) are uncertainties related to the chosen climate-modelling procedure evaluated, and (3) are climate-related uncertainties presented in the results, either as a qualitative evaluation or by statistical analysis.

In addition to the two sets of queries above, the field of research for each study has been noted and categorized. Answers to the questions have been summed up and analyzed for all studies and sub-divided into key categories.

Results from the charting of the data are presented in Section 3. The key findings from the analysis are discussed further in Section 4.

### 3. Results

*3.1. General Overview of the Identified Studies*

In all, 132 studies were identified, and a summary of the different study categories is presented in Figure 2. See Appendix A for a detailed list of the included studies and Tables 3 and 4 for a summary of the review findings. As can be seen from Figure 2, the selection is clearly dominated by studies concerning energy demand and HVAC. This is partly an artifact of the chosen search terms, and partly because articles in these two fields are published more frequently compared to the other research categories. Because of the unknown size of the selection bias resulting from the chosen search terms, ratios between the different categories in Figure 2 cannot be taken as the ratio of publishing frequency between the fields. Approximately two thirds of the identified studies were within the fields of energy demand ($n = 49$) and indoor climate evaluation ($n = 38$). The remaining third was split between performance evaluations of materials and building parts ($n = 15$), sustainability evaluations on a building level or above ($n = 21$), and studies estimating future climate loads for buildings ($n = 9$).

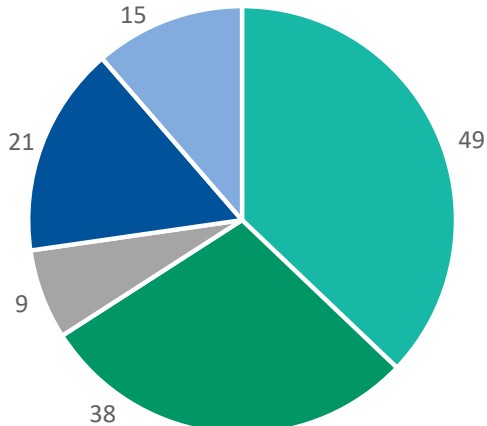

**Figure 2.** Categories of study in the included material (see Table 3 and Appendix A for identification of studies in each category).

**Table 3.** Categorization of climate models used in the study selection, and identification of studies included in each category (see Appendix A for study ID references).

| | Categorization | # | Study ID (See Appendix A for Study Details) |
|---|---|---|---|
| Field of research | Energy demand for heating and cooling | 49 | 3, 4, 6, 12, 14, 15, 18, 21–24, 29, 31–33, 35, 36, 38, 41, 42, 44, 48, 50–54, 56, 65, 67, 71, 73, 76, 77, 80, 82–84, 86, 90, 94, 101, 110, 113, 120, 123, 126, 130, 132 |
| | HVAC, overheating, and indoor climate | 38 | 1, 2, 5, 7, 9, 10, 13, 17, 19, 20, 28, 30, 37, 40, 55, 57, 59, 63, 66, 68–70, 74, 81, 85, 87, 92, 97, 100, 102, 112, 116, 121, 122, 124, 127, 129, 131 |
| | Future climate loads for buildings | 9 | 72, 75, 79, 104–106, 115, 125, 128 |
| | Performance/sustainability (system level) | 21 | 11, 16, 27, 39, 43, 45, 47, 49, 58, 60, 62, 64, 88, 89, 91, 95, 98, 103, 107, 109, 119 |
| | Performance/sustainability (material or component level) | 15 | 8, 25, 26, 34, 46, 61, 78, 93, 96, 99, 108, 111, 114, 117, 118 |
| Climate parameters | Temperature and solar radiation | 130 | 1–98, 100–124, 126–132 |
| | Athmospheric conditions (humidity, $CO_2$-concentrations, pollutants) | 72 | 1–6, 8, 12, 13, 21, 23, 25–28, 30, 33, 37, 38, 42, 44, 46–48, 52, 54, 55, 57, 60, 61, 65, 67, 70, 76–78, 80–85, 87, 89–92, 94–97, 99, 103, 104, 107–109, 112, 114–118, 121, 122, 126–128, 130–132 |
| | Precipitation | 20 | 25, 46, 60, 61, 72, 75, 77, 79, 82, 93, 104, 105, 108, 111, 114, 115, 117, 118, 128, 131 |
| | Wind direction and wind speed | 45 | 1, 2, 5, 6, 12, 13, 25, 27, 28, 30, 33, 47, 48, 54, 57, 60, 61, 65, 67, 70, 75–77, 79, 80, 82, 84, 85, 87, 89, 90, 92, 94, 95, 97, 105, 109, 112, 114–116, 118, 121, 126, 131 |
| | Frost cycles and snow loads | 5 | 72, 75, 117, 126, 128 |
| Model time interval | Hourly values | 98 | 1, 2, 4–7, 9–12, 17–25, 28–30, 32–34, 36–39, 41, 42, 44, 46–51, 54, 55, 57, 59, 61–67, 69, 70, 74, 76, 79–90, 92–98, 100–104, 107, 109, 111–118, 120–128, 131, 132 |
| | Daily values | 15 | 14–16, 35, 40, 53, 58, 71–73, 77, 78, 91, 106, 129 |
| | Monthly values | 18 | 3, 8, 13, 26, 27, 43, 45, 52, 56, 60, 68, 75, 99, 105, 108, 110, 119, 130 |
| Emission scenario | RCP 2.6/B1 | 14 | 51, 53, 67, 70, 91, 97, 99, 101, 106, 108–110, 117, 119 |
| | RCP 4.5/B2 | 46 | 7–9, 12, 20–23, 29, 30, 32–34, 36, 37, 47, 50, 53, 61, 66–68, 71–73, 75, 77, 80, 81, 84, 90, 91, 93, 97, 99, 101, 102, 106, 107, 109, 117, 120, 123, 125, 126, 132 |
| | RCP 6.0/A1B | 30 | 1, 2, 6, 7, 10, 12, 14, 16, 20–22, 24, 25, 27, 29, 32, 34, 36, 37, 39, 40, 51, 57, 58, 70, 74, 87, 94, 109, 130 |
| | RCP 8.5/A1F1/A2 | 90 | 3–5, 7, 8, 12, 15, 17–21, 26, 28, 29, 32–39, 42–48, 50, 51, 53, 56, 59, 63–69, 71, 72, 75, 77, 79–86, 88–95, 97–99, 101, 103, 104, 107–111, 113, 115–124, 126–129, 131, 132 |
| | Non-IPCC scenario (i.e., +T temp. change) | 7 | 52, 55, 61, 62, 105, 112, 114 |
| | Historical extreme | 2 | 11, 49 |
| | Not specified | 9 | 13, 31, 41, 54, 60, 76, 78, 96, 100 |

**Table 3.** *Cont.*

| | Categorization | # | Study ID (See Appendix A for Study Details) |
|---|---|---|---|
| Climate model to weather data | Dynamic downscaling, multiple GCM-RCM chains | 23 | 5, 16, 24, 25, 33, 35, 36, 41, 61, 72, 75, 98, 101, 103–106, 109, 111, 114, 115, 118, 122 |
| | Dynamic downscaling, single GCM-RCM chain | 16 | 1, 2, 6, 14, 23, 28, 30, 31, 34, 48, 57, 66, 81, 85, 117, 125 |
| | Statistical downscaling/morphing, multiple GCM/RCM | 23 | 8, 9, 18, 21, 37, 53, 56, 67, 79, 80, 82, 86, 90, 93, 97, 107, 108, 110, 124, 126, 128, 131, 132 |
| | Statistical downscaling/morphing, single GCM/RCM | 45 | 4, 7, 10, 12, 13, 17, 20, 22, 26, 27, 29, 32, 38–40, 42, 44, 46, 47, 51, 54, 58–60, 62–65, 76, 83, 84, 88, 89, 91, 92, 94–96, 100, 112, 113, 116, 120, 121, 129 |
| | Other | 7 | 11, 15, 43, 49, 70, 71, 78 |
| | Not described | 18 | 3, 19, 45, 50, 52, 55, 68, 69, 73, 74, 77, 87, 99, 102, 119, 123, 127, 130 |
| Result presentation | Probabilistic output (evaluation of climatic uncertainties and/or variance) | 27 | 3, 7, 10, 20, 21, 24, 25, 32, 40, 41, 53, 58, 59, 61, 79, 90, 93, 103, 104, 106, 107, 110, 114, 115, 118, 126, 128 |
| | Deterministic multiple output (alternative climatic outcomes) | 44 | 8, 12, 19, 29, 30, 33–37, 39, 47, 50–52, 55, 65–68, 70–72, 75, 80–82, 84, 91, 94, 97, 98, 101, 105, 108, 109, 111, 112, 117, 120, 122, 123, 131, 132 |
| | Deterministic single output (no alternative climatic outcomes) | 60 | 1, 2, 4–6, 9, 11, 13–18, 22, 23, 26–28, 31, 38, 42, 44–46, 48, 49, 54, 56, 57, 60, 62–64, 69, 73, 74, 76–78, 83, 85–89, 92, 95, 96, 99, 100, 102, 113, 116, 119, 121, 124, 125, 127, 129, 130 |

**Table 4.** Description of future climate uncertainty treatment in the study selection, and identification of studies included in each category (see Appendix A for study ID references).

| | Description | # | Study ID (See Appendix A for Study Details) |
|---|---|---|---|
| Emission scenario | Numerical analysis of scenario variations | 24 | 4, 12, 14, 15, 17, 20, 36, 39, 50, 59, 64, 65, 83, 90, 93, 101, 105, 107, 108, 110, 114, 117, 123, 130 |
| | Description of scenario uncertainty/variation | 73 | 5, 8, 11, 19, 21, 22, 25–30, 32–34, 37, 40–43, 46, 47, 49, 51–53,55, 58, 61, 62, 67, 68, 70–72, 74, 75, 77–81, 84–89, 91, 92, 94, 95, 97–99, 103, 104, 106, 109, 111, 113, 115, 116, 120–122, 124–129, 132 |
| | No description of scenario variations or scenario related uncertainties | 34 | 1–3, 6, 7, 9, 10, 13, 16, 18, 23, 24, 31, 35, 38, 44, 45, 48, 54, 56, 57, 60, 63, 66, 69, 73, 76, 82, 96, 100, 102, 112, 119, 131 |
| Climate model to weather data | Calculated climate model uncertainties or variations | 36 | 3, 6, 7, 9, 10, 20, 21, 24, 25, 29, 30, 32, 41, 58, 59, 61, 72, 75, 78, 79, 81, 90, 93, 103, 104, 106, 107, 109–111, 114, 115, 120, 126, 128, 132 |
| | Description of climate model uncertainties and biases | 30 | 16, 23, 35, 37, 40, 44, 46, 47, 51, 53, 57, 65, 67, 80, 88, 89, 91, 92, 94, 95, 97, 98, 100, 108, 113, 117, 121, 122, 124, 129 |
| | No description of climate model uncertainties and biases | 62 | 1, 2, 4, 5, 11–15, 17–19, 22, 26–28, 31, 33, 34, 36, 38, 39, 42, 43, 45, 48–50, 52, 54–56, 60, 62–64, 66, 68–71, 73, 74, 76, 77, 82–87, 96, 99, 101, 102, 105, 112, 116, 119, 123, 125, 130 |
| Result presentation | Calculation of climate model uncertainties or variations | 30 | 3, 7, 10, 20, 21, 24, 25, 32, 39, 41, 53, 58, 59, 61, 72, 79, 90, 93, 103, 106, 107, 109–111, 114, 115, 118, 122, 126, 128 |
| | Evaluation of climate model uncertainty as an unknown factor | 17 | 4, 15, 23, 29, 30, 40, 67, 75, 80, 81, 88, 97, 104, 105, 116, 121, 132 |
| | Explicit acknowledgement of climate model uncertainty with no evaluation | 29 | 8, 16, 17, 28, 33, 36, 38, 42, 43, 46, 47, 51, 62, 66, 68, 71, 77, 78, 83, 94, 102, 108, 112, 113, 117, 120, 123, 125, 127 |
| | No mention of climate model uncertainty | 56 | 1, 2, 5, 6, 9, 11–14, 18, 19, 22, 26, 27, 31, 34, 35, 37, 44, 45, 48–50, 52, 54–57, 60, 63–65, 69, 70, 73, 74, 76, 82, 84–87, 89, 91, 92, 95, 96, 98–101, 119, 124, 129–131 |

Climate parameters for studies of future energy demand and HVAC performance are mainly influenced by temperature forecasts, which means that the introduced uncertainty is almost exclusively confined to the temperature dimension. This fact may also cause these fields to employ future climate scenarios more frequently when compared to studies using, e.g., models of future precipitation. Studies concerning performance or sustainability of buildings against climate loading need to produce accurate models of multiple climate parameters, two examples being precipitation in combination with temperature and subzero temperatures in combination with recent rainfall. Such multidimensional climate models place a high demand on the method for procuring future climate projections in order to reduce the resulting uncertainties to an acceptable level. The same is true for the studies estimating future climate loads for buildings, i.e., maps of wind-driven rain or future snowfall.

Studies of future energy demand for heating and cooling most frequently comprised one or more case studies of specific buildings. Evaluation of performance after specific retrofitting measures was also commonly found in this category. Some studies looked at energy performance on a district level or higher, pointing out trends on a larger scale. However, the vast majority of studies in this category were case studies of future energy demands, sometimes evaluating other relevant factors in unison, e.g., LCA or cost.

For the studies in the category HVAC, overheating and indoor climate can be largely divided into three categories. The first, and most frequent, was changes in future cooling loads (energy point of view), the second, changes in internal temperatures and overheating risks (comfort point of view), the third, and least frequent, performance of HVAC systems (technical point of view).

The system-level performance category also largely comprised studies looking at adaptation measures for reducing the risk of overheating, e.g., green roofs, thermal mass and U-value of walls, and other passive design strategies. The majority of studies in this category looked at temperature and temperature-related parameters such as solar radiation and relative humidity, as was the case for the two previously described categories.

The majority of studies in the category of future climate loads for buildings looked at development of precipitation patterns, either rain, wind-driven rain, or snow loads. Two studies looked at the development of extreme temperature events, one study climate loads in general (for service-life predictions), while the rest concerned precipitation in some form.

The last category, performance/sustainability on a material or part level, was the most diverse. A total of 6 out of the 15 studies categorized here looked at moisture safety in walls. The most common materials studied were brick or wood (or a combination of the two) in relation to moisture safety, and some studies looked into the carbonatization of concrete. See Table 3 and Appendix A for the collection of studies in the different categories.

*3.2. Use of Emission Scenarios in the Studies*

Of the 132 studies reviewed, 47 studies calculated results from multiple future scenarios. The remaining 85 used only one scenario in the calculations.

Some studies used the Arctic Climate Impact Assessment Report [38], which relies on the A2 and B2 emission scenarios presented by the SRES. In Figure 3, the studies using SRES-type scenarios and Arctic Climate Impact Assessment scenarios have been translated to the equivalent RCP scenario for comparison, using the translation key in Table 5, in accordance with the definitions from the US Global Change Research Program [13].

A total of 43 out of 47 studies utilizing multiple scenarios used RCP 8.5 or the equivalent SRES scenarios (A1F1 or A2) as one of the alternative scenarios. Usually, this was accompanied with RCP 4.5 or the equivalent SRES scenario (B1). In total, 47 of the 85 single-scenario studies used the most severe emission scenario given by the IPCC directly (RCP 8.5 or SRES A1F1/A2), or the equivalent scenario given by the 2004 Arctic Climate Impact Assessment. A total of 9 of the studies did not specify which emission scenario was used to calculate the projected future climate, all of which belonged to the single-scenario category. Few studies discussed reasons for scenario choices in detail, and most frequently the choices seemed to be decided by the availability of relevant climate data in combination with the comprehensiveness of the chosen study rather than active choice. The exception was studies using only RCP 8.5, where the choice was explicitly justified as a comparison between the worst-case future scenario and the present climate. In general, more discussion around the choice of emission scenario was found in studies of multiple scenarios than single, as the scenario variability was explicitly shown in the results when multiple scenarios were calculated.

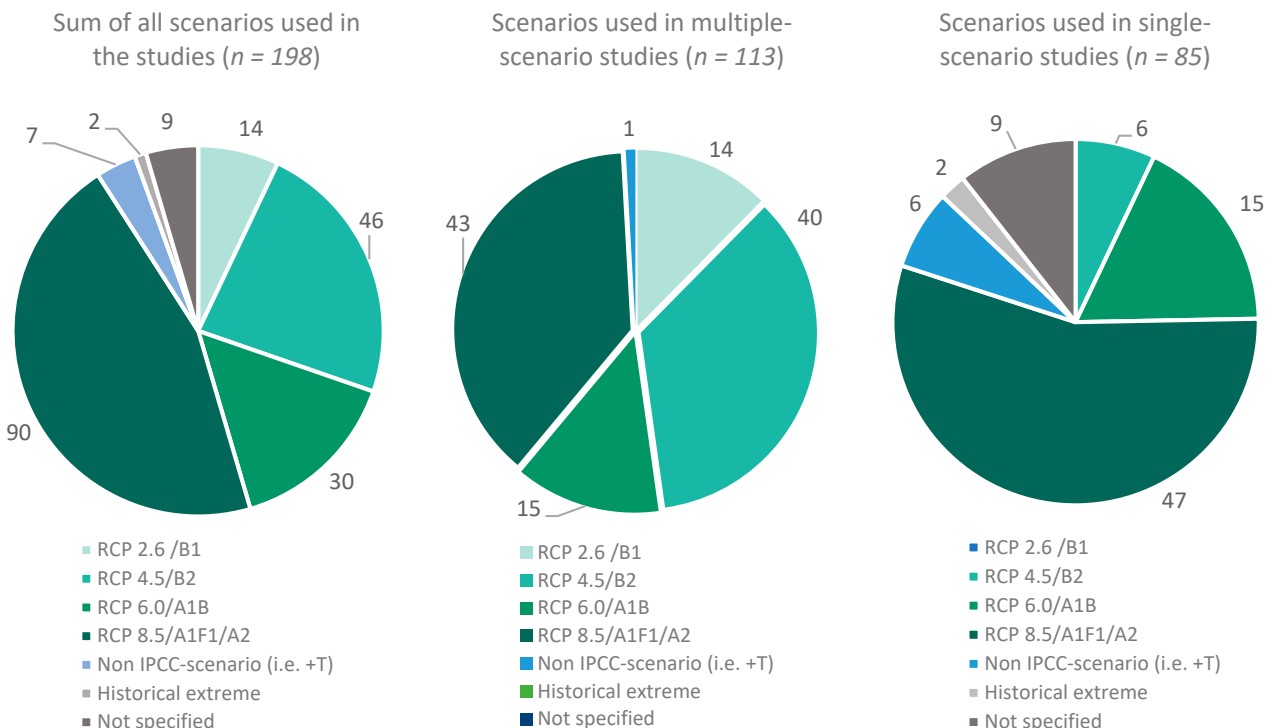

**Figure 3.** Types of emission scenarios used in the studies, divided into studies of multiple scenarios and studies of single scenarios (see Table 3 and Appendix A for identification of studies in each category).

**Table 5.** Comparison between the 2000 Special Report on Emission Scenarios (SRES) and the 2010 Representative Concentration Pathways (RCP).

| SRES Scenario | Equivalent RCP Scenario Emission Prediction |
|---|---|
| A1F1 | RCP 8.5 |
| A2 [a] | RCP 8.5 |
| A1B | RCP 6.0 |
| B2 | RCP 6.0 |
| B1 | RCP 4.5 |

[a] The SRES A2 scenario is a less severe version of the A1F1 scenario, with a predicted atmospheric $CO_2$ concentration in 2100 15% lower than A1F1, but still closer to RCP 8.5 than RCP 6.0.

A total of 17 of the 132 studies only looked at far-future periods (>2070), while the vast majority either looked at near-future (*n* = 42) or development from the present up to far-future (*n* = 68). The remaining 5 studies did not define the future period, but rather defined the scenario as the time where, e.g., $T_{avg}$ = +2 °C. No significant correlations were found between choice of time-frames and emission scenario selection in the analysis of the results.

*3.3. Categorizing the Studies*

For comparison purposes between different characteristics of the studies, the remainder of the review results will be displayed by dividing the studies into 11 subcategories. The subcategories have been chosen based on methodology for producing future climate models and fields of study. The subcategories, and number of studies in each subcategory, is displayed in Figure 4.

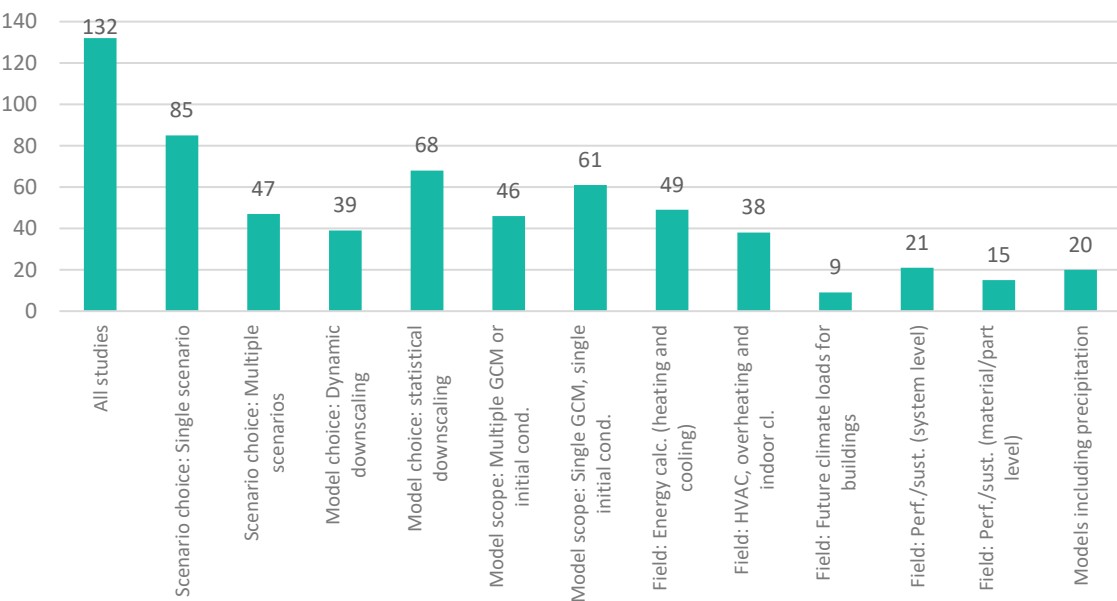

**Figure 4.** Chosen subcategories for the studies, and the number of studies in each subcategory (see Table 3 and Appendix A for the identification of studies in each category).

### 3.4. Methods for Future Climate Data Generation in the Studies

Two principal ways of creating future climate data have been identified in the studies: (1) dynamical downscaling of GCMs to RCMs (*n* = 39), and (2) statistical downscaling of GCMs directly to local weather files (*n* = 68) by determining the relationship between historical global weather patterns and local measurements. Of the remaining studies, 7 used other methods (i.e., employed the use of a historical extreme year as a substitute for future climate), and 18 studies did not specify how the future climate model was created. An overview of the methods used is presented in Figure 5.

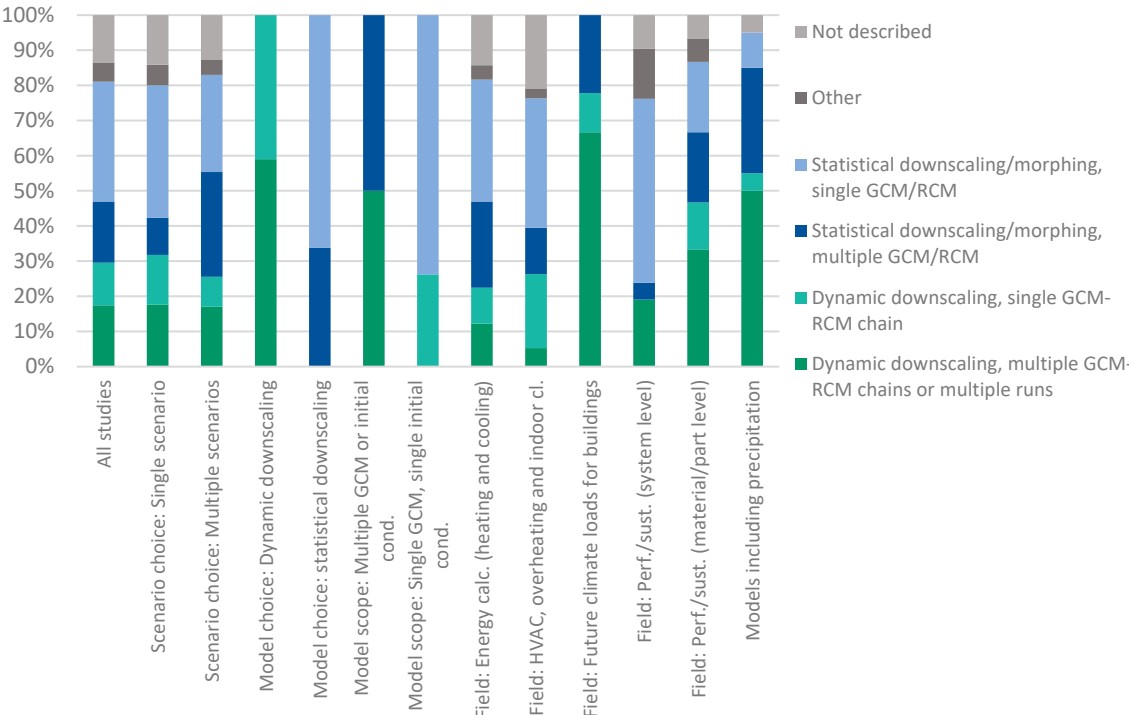

**Figure 5.** Methods for generating future climate models from GCMs (see Table 3 and Appendix A for the identification of studies in each category).

The computational work in statistical downscaling is low compared to dynamic downscaling [17]. For the areas where comprehensive regional models are developed and readily available, there is no need to produce a new model when needed in a new study, and reviewing the studies revealed a close link between certain geographical locations, e.g., Canada, where comprehensible large-scale RCMs are available, and the chosen downscaling method. Statistical downscaling from one GCM is the easiest way to produce local weather data, and indeed 45 of the 68 studies that used statistical downscaling used only one GCM. The ratio of statistical to dynamical downscaling found in this study selection is consistent with findings from other studies, such as Moazami et al., who found that 42 of 111 studies used either dynamical or hybrid downscaling methods [39]. Given that this method is the least time-consuming and most attainable, it is likely that a high number of the studies not specifying the method for climate modelling also used statistical downscaling from one GCM. The availability of high-quality local models is therefore paramount to determine and reduce uncertainties from climatic input values in climate adaptation research.

### 3.5. Uncertainty Assessments of Climate Model Influence on Output Values

To answer research question 2 (see Section 1.1), the studies have been sorted by how uncertainties in the climate model are treated, and how they are presented in the results. All studies have been characterized by uncertainty treatment in 3 steps: (1) treatment of emission scenario uncertainties, (2) treatment of climate model uncertainty, and (3) presentation of result uncertainty with regard to future climate estimation. The results are presented in Figures 6–8.

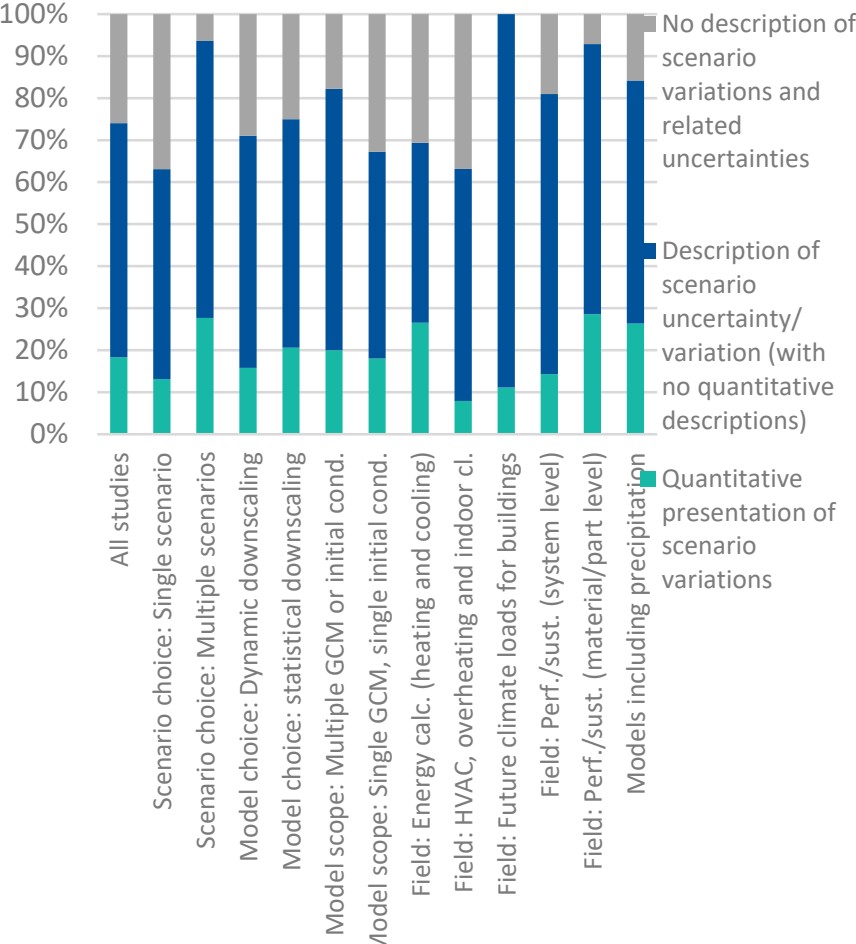

**Figure 6.** Emission scenario uncertainty evaluation in method description (see Table 4 and Appendix A for identification of the studies in each category).

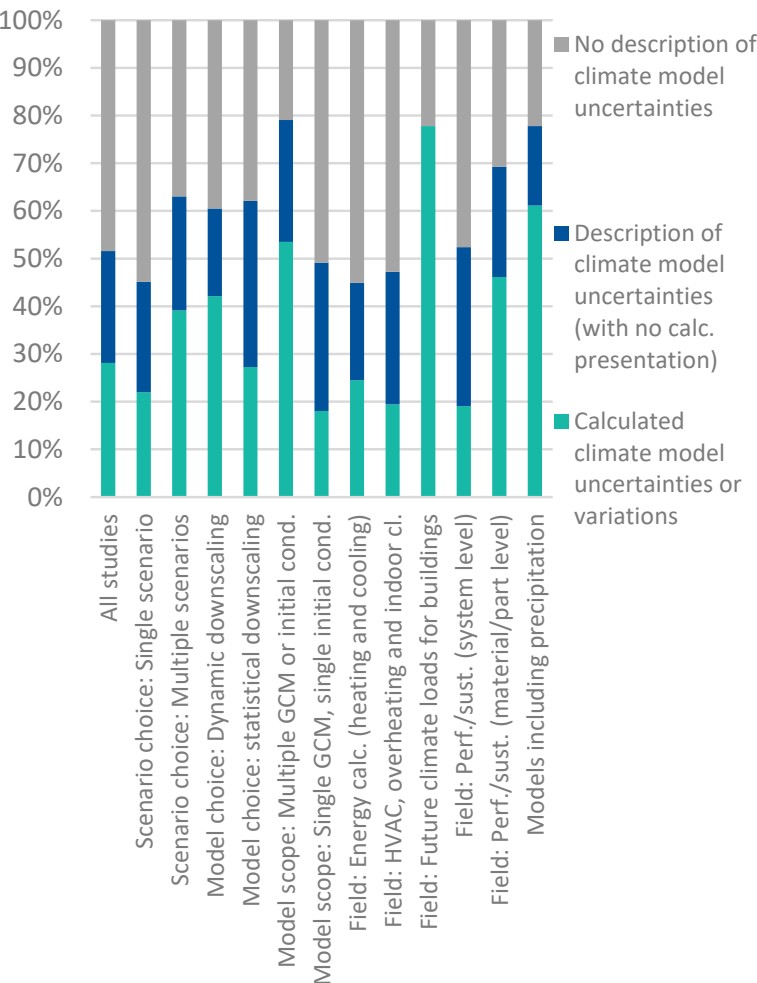

**Figure 7.** Climate model uncertainty evaluation in method description (see Table 4 and Appendix A for identification of the studies in each category).

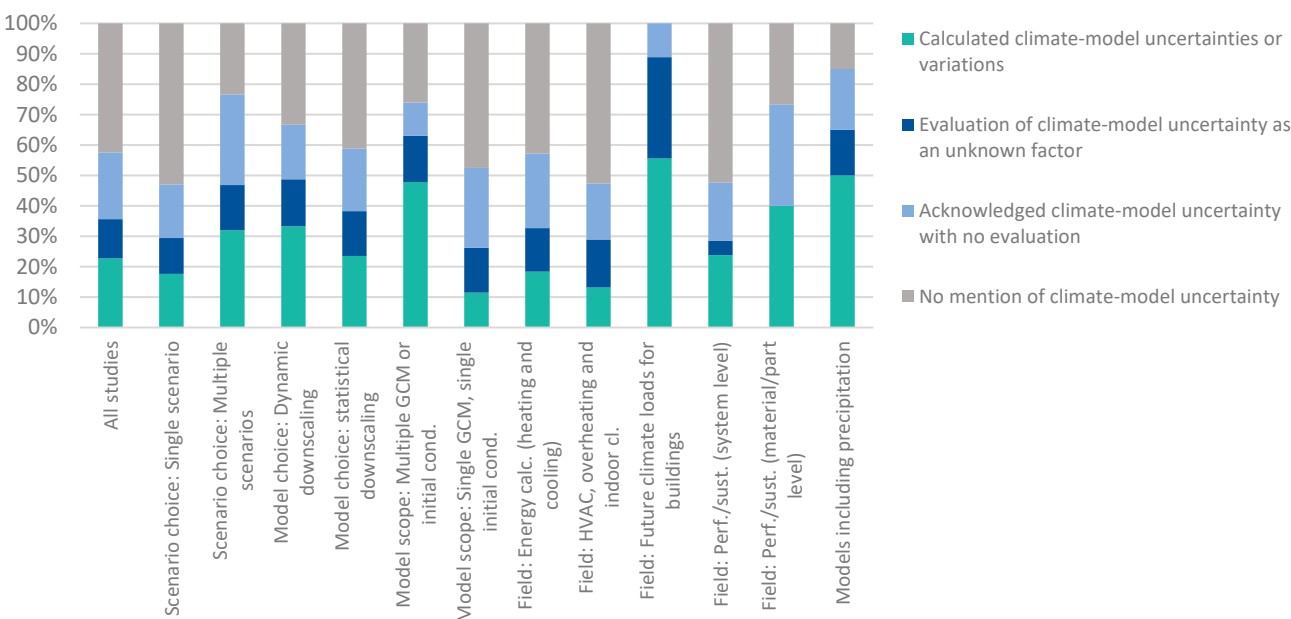

**Figure 8.** Presentation of emission-scenario- and climate-model-induced uncertainty in study results, discussion, and conclusions (see Table 4 and Appendix A for identification of the studies in each category).

In total, 30 out of 132 studies quantified some form of future-climate-related uncertainty as a part of the climate model output and integrated it into the results. A further 17 studies acknowledged the climate-model-induced uncertainties and discussed how they might impact the results by qualitative analysis. In total, 29 studies acknowledged climate-model-induced uncertainties with no further evaluation, and 56 studies did not mention either climate-model-induced uncertainties or scenario choice as a source of uncertainty at all in the presentation and discussion of the results and the following conclusions.

The studies that evaluated climate model uncertainty as an unknown factor frequently focused most on evaluating the scenario selection, typically calculating a worst-case scenario followed by a discussion of how other scenarios might impact the results or by discussing results from two or more calculated scenarios and evaluating the differences. Evaluation of uncertainties induced by climate model choices also occurred but less frequently, i.e., by evaluation of downscaling method, GCM selections, bias corrections, or weather file generation.

Of the studies that go deeply into uncertainties, by including some form of uncertainty calculation as a part of the results, most of them conclude that analysis and estimation of the uncertainties in the climate model should be made. Further, they conclude that the analyzed uncertainties have a significant impact on the results, connected to variance in simulation output in producing weather files from GCMs/RCMs and the effect of subsampling GCMs, and they do not recommend omitting this from the results as it could lead to misleading conclusions.

### 3.6. Characterizing and Mitigating Uncertainties

Uncertainty comes in many different flavors, as the hierarchy proposed by Riesch et al. illustrates [6]. The uncertainty level hierarchy proposed by Riesch et al. is an excellent tool to break up and characterize a set of uncertainties in a given model, and the type classification of uncertainties in climate modelling provided by Ekstrøm et al. [19] (see Section 1.4) provides an excellent way to break up the different models required to make future climate predictions. The mitigation methods for treating uncertainties related to emission scenarios, climate models, and natural variability of weather patterns found in the 132 studies are presented in Table 6, using Riesch's method for uncertainty characterization and Ekstrøm's classification of uncertainty types in future climate predictions. For a more detailed description of the five uncertainty categories in the columns of Table 6, see Table 1.

Table 6 attempts to summarize the different strategies used for uncertainty mitigation. Only mitigating measures found to be used in the studies are listed, and the table may be viewed as a checklist with suggestions for mitigating measures with regard to final uncertainty levels in the results.

The methodological steps necessary to mitigate all the different uncertainties presented in Table 6 include selecting multiple scenarios, using multiple climate models, and calculating multiple final result outputs for statistical treatment, such as expressions of means and spreads. In total, 11 of the studies in the selection (21, 53, 61, 72, 90, 93, 106, 107, 109, 110, 126 in Appendix A) were found to demonstrate this (see Figure 9). These studies were characterized by having a comprehensive selection of GCMs/RCMs at their disposal, often producing separate climate projections from each one. This enabled them to evaluate an ensemble of climate outputs from different GCMs, thus carrying the GCM variance into the final results of, e.g., the energy calculations. Some of the 11 studies were large-scale studies of, e.g., future trends of energy use in Europe, using an ensemble of RCMs already gathered through the EURO-CORDEX program [15], thus avoiding downscaling and creating hourly weather file generation altogether. Morphing multiple types of weather years, e.g., typical, extreme-low, and extreme-high, using Finkelstein–Schafer statistics was a frequent method used to mitigate the uncertainties from natural variation (type 3).

**Table 6.** Characterization of principal uncertainties found in climate modelling for building adaptation measure purposes, and methods for mitigation. Mitigation measures are based on findings in the study selection and are confined to measures controllable by methodological choices in building adaptation measure research.

| | | Level 1 (Uncertainty about the Outcome) | Level 2 (Uncertainty about the Parameters) | Level 3 (Uncertainty about the Model Choice) | Level 4 (Uncertainty about Acknowledged Inadequacies in the Chosen Model) | Level 5 (Uncertainty about Unknown Inadequacies) |
|---|---|---|---|---|---|---|
| Emission scenario uncertainty (type 1) | characterization | | Non-exhaustive empirical climate information | Validity of emission scenarios | Complexity of global climate systems, effects of high-impact parameters | Future socio-economic conditions and unknown climate effects |
| | mitigation | | N/A | Calculation of multiple scenarios | Evaluation of selected scenario validity | N/A |
| Climate model uncertainty (type 2) | characterization | | Downscaling precision level, topographic influence | GCM variability, choice of downscaling method | Non-perfect climate modelling and weather file generation | Unknown high-impact parameters influenced by increased emission levels. |
| | mitigation | | Calibration of the model to empirical data, bias correction | Calculation of multiple RCM-GCM chains or statistical downscaling of GCM ensemble. | Evaluation of inadequacies in the chosen climate modelling method | N/A |
| Result model uncertainty (type 3) | characterization | Natural variability in stochastic generation of weather files | Non-perfect input from climate model and stochastic weather files | Variability between emission scenarios | Total accumulated uncertainty from emission scenario to result | |
| | mitigation | Generation of multiple weather files for statistical analysis | Calculation of the climate model's influence on result outcome space | Calculation of results from multiple scenarios | Evaluation of result validity | |

The comprehensive methods discussed above for uncertainty treatment were presented with expressions of spreads, standard deviations, or variance in addition to an average value in the results. Studies that did this often discussed the different sources of uncertainties that created the result variance, and conclusions were made after careful consideration of the climate projection validity.

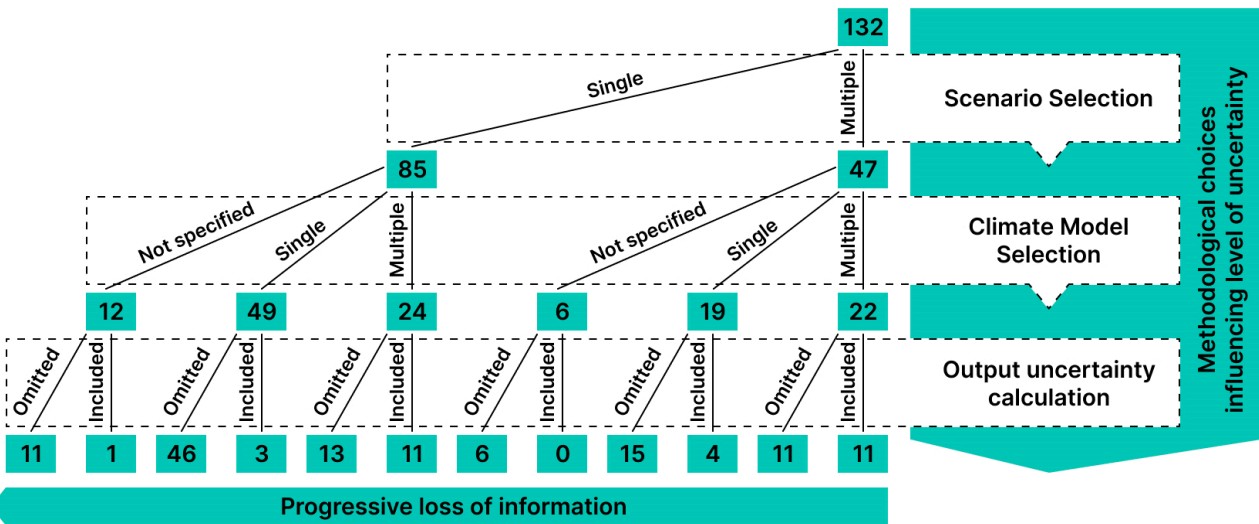

**Figure 9.** Stepwise loss of information about climate-model-induced uncertainties in results, depending on methodological choices for future climate modelling. The numbers in each square represent the number of reviewed studies found to follow a particular path.

## 4. Discussion

Climate model comprehensiveness varies widely between studies, and ranges from bias-corrected dynamic downscaling of an ensemble of GCMs forced by multiple emission scenarios to a statistical downscaling of a single GCM from one emission scenario. Quantification and assessment of model uncertainties is just as varied, although comprehensive climate models tend to correlate with a comprehensive analysis of uncertainties. In the following sections, a deeper dive into the relevant findings for the two research questions stated in Section 1.1 is presented, summing up with a proposed hierarchical description of climate model prediction uncertainties based on Ekstrøm et al.'s three climate model uncertainty types [19] as a tool for assessing the type and comprehensiveness of climate-model-induced uncertainties accumulated in the results.

### 4.1. Selecting Emission Scenarios

More than half of the studies that used only one emission scenario used RCP 8.5 or the IPCC AR4 equivalents as the scenario of choice. Arguably, the choice of emission scenario may have the greatest influence on the resulting future climate estimations, given longer timeframes (i.e., 2071–2100). The results show that 85 of the 132 studies reviewed either looked at future conditions in 2071–2100 or developments towards 2100 (see Section 3.2). The difference in precipitation and temperature predictions between RCP 4.5 and RCP 8.5 has been shown to be larger than the variation between different GCMs for considerations of 2071–2100 climate [30]. A study comparing different sources of uncertainties in climate modelling looked at how different initial conditions, different choice of GCM, and different spatial resolutions influenced rain intensities and found that these factors may induce 10–40% variance in the rain intensity output (30-year average and standard deviation) [40]. The choice of GCM was the largest source of uncertainty, given emission scenario A1B [40]. The same study found that different emission scenarios affected rain intensity uncertainty levels by as much as 80%.

Compared to mapping the parameter variations for different GCMs, mapping parameter variations for different scenarios requires considerably less effort. In total, 47 out of 132 studies calculated multiple emission scenarios; 22 of these included multiple GCM inputs in their model, although only 11 used this to calculate the result variation caused by GCMs (see Figure 9). The most common pathway in the study selection was models using a single emission scenario (*n* = 85) with a single GCM input (*n* = 49), presenting no climate-induced variations in the results (*n* = 46). By choosing only one scenario, the results

from the study will be uncertain by design, and the choice of scenario should therefore be considered carefully. If the study must be limited to one scenario because of, e.g., financial constraints, it makes sense to use the most extreme scenario when compared to the situation of today as most of the single-scenario studies in this review have done. In such a study, today's climate will constitute one outer extreme, and the most severe future scenario will constitute the other outer extreme. Of course, global emissions may turn out to exceed the RCP 8.5 scenario, but RCP 8.5 is the most severe scenario presented by the IPCC in AR5, and as such it may be viewed as a representative worst-case scenario. The advantage of knowing today's situation and the most severe future scenario is that the development direction is known. If the results of the research are to be used in building design or maintenance, the reader can then evaluate how their given situation relates to the two extremes. However, one important condition must be met for this method to be used: some way of interpolating values between the two extremes must be available to the reader if intermediate results are to be considered. Complex parameters such as rain intensity or frost cycles may not easily lend themselves to such an approach, as the responses of both to global climate changes have high uncertainties connected to them and they behave in nonlinear ways.

### 4.2. Determining, Presenting, and Evaluating Uncertainties

There are two principal sources of uncertainties connected to future climate assessments, in addition to natural variability caused by weather patterns. The first is connected to the emission scenario itself, as determining future emission levels requires assumptions of socio-economic character. The second is climatic response uncertainty due to imperfect knowledge and models. Previous works have shown that these two different types of uncertainties may be confounded in the final results [41], and expressions of uncertainty are often attributed to the climate models only, ignoring scenario-related uncertainties. By displaying results from only one emission scenario, this type of uncertainty is hidden from view. As illustrated in Figure 3, more than half the studies using only one scenario used RCP 8.5. If the uncertainties in the emission scenario are ignored, variations in future climate due to emission intensity are ignored. This may in turn lead to a falsely deterministic view of the future. A further nine of the studies using only one scenario did not specify which scenario was used and did not express any form of uncertainty in the results and evaluations. Failure to evaluate the uncertainties altogether gives a false perception of certainty, and failure to report what type of future scenario the results are based on leaves the reader helpless to evaluate for themselves.

The studies presenting the most comprehensive evaluation of climate-related uncertainties produced means and standard deviations for all used GCM-RCM chains and evaluated the variations. The methods for downscaling and bias correcting the climate models were well-documented in these studies, providing a transparent presentation of the underlying uncertainties. Producing such results is demanding work and probabilistic results may be more challenging to present in a clear way. However, presenting a comprehensive analysis of uncertainties greatly increases the quality of the results, and even more importantly, it reminds the reader that assessments of complex systems far into the future are not deterministic in nature.

Some uncertainties are easier to mitigate than others, and others again might be impossible to mitigate due to the complexity of global climate modelling and unforeseen future socio-economic developments. These higher-level uncertainties have been described by researchers within the field of risk and uncertainty as unquantifiable, irreducible epistemological limits related to inductive reasoning and to the nature of model-based global science [7]. However, lower-level uncertainties (see Table 1) may be mitigated either through development of better models or through methodological choices in a particular study (see Table 6).

### 4.3. A proposed Hierarchy of Uncertainty in Climate Modelling

Uncertainties in climate modelling are connected to the whole chain, from emission scenario to study-specific output. Many studies take uncertainties from one or more of these links into consideration, but few studies attempt to quantify the total uncertainty involved. Finding an expression for such a "total uncertainty" in the results is demanding work, impractical, costly, and arguably ultimately impossible [7], as discussed in a guidance note by the IPCC on uncertainty treatment in climate modelling [35]. However, there are tools and methods available for calculating the quantifiable sources of uncertainties in an adequate way, and yet other tools and methods for qualitatively analyzing the rest will increase the reliability of the results, as most of the studies in the selection using such tools and methods pointed out. In particular, (1) calculating different emission scenarios and (2) analyzing variance in simulation output for weather files when downscaling from GCM/RCM were frequently identified as important mitigation measures for large sources of uncertainty [42–44]. Further mitigation measures may be implemented by increasing the model quality, i.e., through bias corrections or by increasing the model fit, i.e., through selecting an appropriate spatial resolution in dynamic downscaling.

The first two of the three principal types of uncertainty in future climate assessments identified by Ekstrøm et al. [19] (see Section 1.4), scenario uncertainty and climate model uncertainty, can be ranked hierarchically. The climate model output depends on the emission scenario, and as previously shown, the scenario has a greater influence on output values than variations in output between established GCMs [33]. Still, GCM uncertainties have been shown to be large as well, as the example of temperature variation illustrated in Figure 1 shows. Output values of parameters such as precipitation levels vary even more significantly between GCMs [33], making use of multiple GCMs even more important for studies of moisture safety design, etc. A dynamic downscaling of a GCM to RCM will carry the biases of the GCM and add uncertainty from other sources as well. Using multiple GCM-RCM chains and proper bias correction will reduce this source of uncertainty. By displaying the sources of climate-related uncertainties discussed in this paper in a stepwise fashion, a principal hierarchy of uncertainty describing the level of final uncertainty can thus be proposed. The hierarchy is presented in Figure 9, and the number of studies choosing the different paths are displayed inside the boxes.

The final level of uncertainty in the chain displayed in Figure 9 is related to the presentation and assessment of the results. Including multiple outputs in the results illustrates variance, and thus the uncertain nature of future climate predictions. The third principal type of uncertainty identified by Ekstrøm et al. [19], natural variability in weather patterns due to the chaotic nature of the climate system, may only be evaluated through multiple outputs from the models, i.e., stochastic generation of multiple weather files. The treatment quality of this type of uncertainty will be determined by the quality of uncertainty treatment in the two upper levels of Figure 9. The studies that calculated expressions of result uncertainties frequently communicated them using averages and spreads (e.g., [43,45–47]). Other strategies included reporting variance and confidence intervals through the creation of an ensemble of weather files. More often than not, these studies concluded that such calculations should not be omitted. Approximately half of the studies using multiple climate models calculated an expression of uncertainty in the results (22 out of 46), whereas less than 10% of the remaining studies were able to do the same (8 out of 86).

When calculating expressions of result uncertainty, it is equally important to communicate what type of uncertainties are included in the calculations, as some methods reduce uncertainties through multiple weather file creations (type 3), and some through the use of multiple climate models (type 2). In total, 56 studies did not communicate any uncertainties associated with future climate estimations at all (see Table 4). By omitting information about the aggregated uncertainties from the chosen scenario and the model used, the reader of the study will have no way of judging the reliability of the results. This type of uncertainty is the only one that can be eliminated in full. By careful description of

the methods used to produce the models and by evaluating the validity of the results, no information is lost through communication failure.

The proposed hierarchy in Figure 9 does not differentiate between dynamic and statistical downscaling approaches of global data. The uncertainties connected to the two approaches are qualitatively different, but the quantified uncertainty level in the model outputs will depend more on the quality of the downscaling than the type of approach. Dynamic downscaling yields more precision relative to a GCM, but statistical downscaling of multiple GCMs may in turn yield more information about the uncertainties and biases induced by the particular GCMs. Multiple GCMs can be statistically downscaled and compared with less effort than by the dynamic downscaling of a single GCM, thus revealing the result variation induced by the GCM particularities. Dynamic downscaling from an ensemble of GCMs or a combination of dynamic downscaling and multiple statistical downscaling may be the best option for controlling model-induced uncertainties, albeit also the most comprehensive and time consuming. If this option is unavailable due to, e.g., economical constraints or lack of model availability, evaluating the unquantified uncertainties in the results will at least increase result credibility and usability.

### 4.4. Future Needs for Tools in Building Climate Adaptation Research

Regarding treatment and communication of uncertainties in climate modelling, there are two main points found in this study. The first is that awareness of uncertainty induced by future climate estimation should be improved in building adaptation research. The second is that, although methods and tools for estimating and mitigating uncertainties exist, they are not yet widespread enough for common use.

Climate-model-induced uncertainties were completely neglected when estimating future climate impacts on building adaptation measures in 56 out of 132 of the studies in this selection. This may simply be due to lack of awareness in many cases, as many studies treated model output from synthesized future weather files in the same way as model output from historical measurements, with no mention of future climate estimation as a source of uncertainty in the calculations. In total, 46 studies treated this source of uncertainty through qualitative evaluation, but they suffered from a lack of uncertainty expressions in the model outputs. Whether they lacked the necessary quality of data, or they lacked the necessary tools to implement them, these studies did not employ any methods for mitigating or estimating the uncertainties induced by the models through calculations. This points to a central fact, namely that researchers within the field of building adaptation research are not meteorologists. As future climate modelling is complex, it is difficult to estimate, and even be aware of, the different sources of uncertainties introduced by the future climate models. The availability of high-quality climate models with probabilistic or statistical descriptions of uncertainty is therefore paramount. The studies of areas where such models exist, such as the UK and Canada, were found to have a more explicit relationship to climate-model-induced uncertainties than studies of other areas (e.g., [42,47,48]).

A building physicist researching moisture problems for a solution, or a ventilation researcher studying cooling demands in indoor climates, may not have the expertise to know the inner workings of the climate model they are using as their tool. Likewise, the meteorologist making the future climate model may not be aware of the building adaptation researchers' needs for uncertainty expressions, or which type of synthesized weather files are most appropriate for the problem at hand. Thus, awareness and readily available tools for treating uncertainties are paramount for both researchers. As briefly presented in Sections 1.3 and 1.4, there are many excellent works in the literature looking at methods for handling the technical challenges pertaining to future climate modelling for use in building adaptation research, both for estimation and for mitigation of different sources of uncertainty. Making such methods standard use, and finding good ways of communicating the uncertainty evaluation, would enhance the quality and reliability of the results in building climate adaptation research.

Ekstrøm et al.'s differentiation of climate prediction uncertainty types [19] provides an excellent basis for both the treatment and communication of uncertainties in future climate calculations. By breaking up the uncertainties into three distinct sources; emission scenarios, climate models, and natural variability in weather patterns; the types of uncertainties may be treated separately. Discussing and communicating both treated and untreated uncertainties then follows naturally.

## 5. Conclusions

The study selection comprised 132 research papers from 2014 to 2021, exploring building adaptation measures using future climate scenarios.

The primary methods for producing future climate scenarios in research concerning future climate adaptation measures for buildings are dynamic downscaling and statistical downscaling of GCMs or RCMs. The comprehensiveness of the models varies, but the most-used method in the study selection was statistical downscaling from a single GCM or RCM. RCP 8.5 or the IPCC AR4 equivalent was the most frequently used emission scenario for studies investigating a single scenario. Studies investigating multiple scenarios most frequently compared RCP 8.5 and RCP 4.5 or the IPCC AR4 equivalent. The number of studies in the selection found to be using a single scenario was 85, whereas 47 studies used multiple scenarios.

Climate-model-induced uncertainties influencing the results were evaluated and communicated to a varying extent in the study selection. Uncertainties were neglected more frequently than not. In total, 56 of 132 studies did not mention future climate modelling as a source of uncertainty at all, and a further 29 studies acknowledged future climate modelling as a source of uncertainty without evaluating the influence this might have on the results. Of the 47 studies that did evaluate future-climate-modelling-induced uncertainties, 30 studies calculated an expression of uncertainty and integrated it as part of the results, and the remaining 17 evaluated the uncertainties qualitatively. A frequent conclusion found in the studies analyzing and estimating the uncertainties in the climate models was that this should not be omitted, as it was found to have a great impact on the results.

There is a growing number of studies describing methods and tools for mitigating future-climate-model-induced uncertainties, but the tools are not yet widespread enough. Uncertainties introduced by future climate estimation were in general under-communicated in the study selection, commonly resulting from either lack of awareness or lack of sufficient analysis of the result uncertainties. The availability of high-quality climate models with probabilistic or statistical descriptions of uncertainty is an important prerequisite for communicating uncertainties adequately, as is the awareness of uncertainties in future climate estimation and the tools for mitigating them.

**Author Contributions:** J.E.G.: Conceptualization, methodology, formal analysis, investigation, writing—original draft, visualization, H.O.H.: validation, writing—review and editing, supervision, R.A.B.: methodology, supervision, T.K.: conceptualization, methodology, writing—review and editing, main supervision. All authors have read and agreed to the published version of the manuscript.

**Funding:** This research was funded by the Research Council of Norway, grant number 237859.

**Data Availability Statement:** No new data were created or analyzed in this study. Data sharing is not applicable to this article.

**Acknowledgments:** The authors gratefully acknowledge the financial support from The Research Council of Norway and several partners through the Centre for Research-based Innovation 'Klima 2050' (see www.klima2050.no (accessed on 1 June 2023)).

**Conflicts of Interest:** The authors declare that they have no known competing financial interests or personal relationships that could have influenced the work reported in this paper.

## Appendix A

**Table A1.** List of articles included in the study, with corresponding ID numbers for identification in the analysis results of the main article.

| ID # | Author | Year | Journal | Title |
|------|--------|------|---------|-------|
| 1 | Berger et al. | 2014 | Building and Environment | Impacts of urban location and climate change upon energy demand of office buildings in Vienna, Austria |
| 2 | Berger et al. | 2014 | Energy and Buildings | Impacts of climate change upon cooling and heating energy demand of office buildings in Vienna, Austria |
| 3 | Braun et al. | 2014 | Applied Energy | Using regression analysis to predict the future energy consumption of a supermarket in the UK |
| 4 | Daly et al. | 2014 | Building and Environment | Implications of global warming for comercial building retrofitting in Australian cities |
| 5 | Kalvelage et al. | 2014 | Energy and Buildings | Changing climate: The effects on energy demand and human comfort |
| 6 | Orehounig et al. | 2014 | Sustainable Cities and Society | Projections of design implications on energy performance of future cities: a case study from Vienna |
| 7 | Patidar et al. | 2014 | Renewable Energy | Simple statistical model for complex probabilistic climate projections: Overheating risk and extreme events |
| 8 | Saha et al. | 2014 | Urban Climate | Urban scale mapping of concrete degradation from projected climate change |
| 9 | Sailor | 2014 | Building and Environment | Risks of summertime extreme thermal conditions in buildings as a result of climate change and xacerbation of urban heat islands |
| 10 | Taylor et al. | 2014 | Building and Environment | The relative importance of input weather data for indoor overheating risk assessment in dwellings |
| 11 | van Hooff et al. | 2014 | Building and Environment | On the predicted effectiveness of climate adaptation measures for residential buildings |
| 12 | Wang et al. | 2014 | Energy and Buildings | Impact of climate change heating and cooling energy use in buildings in the United States |
| 13 | Yau et al. | 2014 | Energy and Buildings | The performance study of a split type air conditioning system in the tropics, as affected b weather |
| 14 | Zachariadis et al. | 2014 | Energy | The effect of climate change on electricity needs- a case study from Mediterranean Europe |
| 15 | Zhou et al. | 2014 | Applied Energy | Modeling the effect of climate change on US state-level buildings energy demands in an integrated assessment framework |
| 16 | Andersson-Sköld et al. | 2015 | Climate Risk Management | An integrated method for assessing climate-related risks and adaptation alternatives in urban areas |
| 17 | Barbosa et al. | 2015 | Building and Environment | Chlimate change and thermal comfort in Southern Europe housing: A case study from Lisbon |
| 18 | Dirks et al. | 2015 | Energy | Impacts of climate change on energy consumption and peak demand in buildings: A detailed regional approach |
| 19 | Guan | 2015 | Architectural Science Review | The influence of internal load density on the energy and thermal performance of air-conditioned office buildings in the face of global warming |
| 20 | Jenkins et al. | 2015 | Buildings | Quantifying change in buildings in a future climate and their effect on energy systems |
| 21 | Jylhä et al. | 2015 | Energy and Buildings | Energy demand for the heating and cooling of residential houses in Finland in a changing climate |

**Table A1.** *Cont.*

| ID # | Author | Year | Journal | Title |
|------|--------|------|---------|-------|
| 22 | Karimpour et al. | 2015 | Energy and Buildings | Impact of climate change on the design of energy efficient residential building envelopes |
| 23 | Kikumoto et al. | 2015 | Sustainable Cities and Society | Study on the future weather data considering the global and local climate change for building energy simulation |
| 24 | Nik et al. | 2015 | Energy and Buildings | A statistical method for assessing retrofitting measures of buildings and ranking their robustness against climate change |
| 25 | Nik et al. | 2015 | Building and Environment | Future moisture loads for building facades in Sweden: Climate change and wind-driven rain |
| 26 | Sajjadian et al. | 2015 | Energy and Buildings | The potential of phase change materials to reduce domestic cooling energy loads for current and future UK climates |
| 27 | Virk et al. | 2015 | Energy and Buildings | Microclimatic effects of green and cool roofs in London and their impacts on energy use for a typical office building |
| 28 | Alves et al. | 2016 | Energy and Buildings | Residential buildings' thermal performance and comfort for the elderly under climate changes context in the city of São Paulo, Brazil |
| 29 | Andric et al. | 2016 | Energy and Buildings | Modeling the long-term effect of climate change on building heat demand: Case study on a district level |
| 30 | Arima et al. | 2016 | Energy and Buildings | Effect of climate change on building cooling loads in Tokyo in the summers of the 2030s using dynamically downscaled GCM data |
| 31 | Berger et al. | 2016 | Journal of Building Engineering | Impacts of external insulation and reduced internal heat loads upon energy demand of offices in the context of climate change in Vienna, Austria |
| 32 | Braun et al. | 2016 | Energy and Buildings | Estimating the impact of climate change and local operational procedures on the energy use in several supermarkets throughout Great Britain |
| 33 | Dodoo et al. | 2016 | Energy | Energy use and overheating risk of Swedish multi-storey residential buildings under different climate scenarios |
| 34 | Fontanini et al. | 2016 | Energy and Buildings | Exploring future climate trends on the thermal performance of attics: Part 1–Standard roofs |
| 35 | Huang et al. | 2016 | Climatic Change | Impact of climate change on US building energy demand: sensitivity to spatiotemporal scales, balance point temperature, and population distribution |
| 36 | Huang et al. | 2016 | Energy | The variation of climate change impact on building energy consumption to building type and spatiotemporal scale |
| 37 | Huang et al. | 2016 | Applied Energy | Future trends of residential building cooling energy and passive adaptation measures to counteract climate change: The case of Taiwan |
| 38 | Invidiata et al. | 2016 | Energy and Buildings | Impact of climate change on heating and cooling energy demand in houses in Brazil |
| 39 | Makantasi et al. | 2016 | Advances in Building Energy Research | Adaptation of London's social housing to climate change through retrofit: a holistic evaluation approach |
| 40 | Mulville et al. | 2016 | Building Research and Information | The impact of regulations on overheating risk in dwellings |
| 41 | Nik et al. | 2016 | Energy and Buildings | Effective and robust energy retrofitting measures for future climatic conditions—Reduced heating demand of Swedish households |
| 42 | Pagliano et al. | 2016 | Energy and Buildings | Energy retrofit for a climate resilient child care centre |

**Table A1.** *Cont.*

| ID # | Author | Year | Journal | Title |
|------|--------|------|---------|-------|
| 43 | Perreault et al. | 2016 | Cold Regions Science and Technology | Seasonal thermal insulation to mitigate climate change impacts on foundations in permafrost regions |
| 44 | Rubio-Bellido et al. | 2016 | Energy | Optimization of annual energy demand in office buildings under the influence of climate change in Chile |
| 45 | Santamouris | 2016 | Energy and Buildings | Cooling the buildings–past, present and future |
| 46 | Sehizadeh et al. | 2016 | Building and Environment | Impact of future climates on the durability of typical residential wall assemblies retrofitted to the PassiveHaus for the Eastern Canada region |
| 47 | Shen et al. | 2016 | Energy | Vulnerability to climate change impacts of present renewable energy systems designed for achieving net-zero energy buildings |
| 48 | Shibuya et al. | 2016 | Energy and Buildings | The effect of climate change on office building energy consumption in Japan |
| 49 | van Hooff et al. | 2016 | Energy | Analysis of the predicted effect of passive climate adaptation measures on energy demand for cooling and heating in a residential building |
| 50 | Waddicor et al. | 2016 | Building and Environment | Climate change and building ageing impact on building energy performance and mitigation measures application: A case study in Turin, northern Italy |
| 51 | Andric et al. | 2017 | Energy and Buildings | The impact of climate change on building heat demand in different climate types |
| 52 | Ascione et al. | 2017 | Energy and Buildings | Resilience of robust cost-optimal energy retrofit of buildings to global warming: A multi-stage, multi-objective approach |
| 53 | Damm et al. | 2017 | Climate Services | Impacts of +2 °C global warming on electricity demand in Europe |
| 54 | Fahmy et al. | 2017 | Energy and Buildings | On the green adaptation of urban developments in Egypt; predicting community future energy efficiency using coupled outdoor-indoor simulations |
| 55 | Hamdy et al. | 2017 | Building and Environment | The impact of climate change on the overheating risk in dwellings—A Dutch case study |
| 56 | Huang et al. | 2017 | Energy and Buildings | Impact of climate change on US building energy demand: Financial implications for consumers and energy suppliers |
| 57 | Hwang et al. | 2017 | Energy and Buildings | Spatial and temporal analysis of urban heat island and global warming on residential thermal comfort and cooling energy in Taiwan |
| 58 | Kingsborough et al. | 2017 | Climate Risk Management | Development and appraisal of long-term adaptation pathways for managing heat-risk in London |
| 59 | Liu et al. | 2017 | Building and Environment | High resolution mapping of overheating and mortality risk |
| 60 | Mosoarca et al. | 2017 | Engineering Failure Analysis | Failure analysis of historical buildings due to climate change |
| 61 | Nik | 2017 | Energy and Buildings | Application of typical and extreme weather data sets in the hygrothermal simulation of building components for future climate—A case study for a wooden frame wall |
| 62 | Pierangioli et al. | 2017 | Building Simulation | Effectiveness of passive measures against climate change: Case studies in Central Italy |
| 63 | Rubio-Bellido et al. | 2017 | Building Simulation | Application of adaptive comfort behaviors in Chilean social housing standards under the influence of climate change |

**Table A1.** *Cont.*

| ID # | Author | Year | Journal | Title |
|------|--------|------|---------|-------|
| 64 | Sajjadian | 2017 | Buildings | Performance Evaluation of Well-Insulated Versions of Contemporary Wall Systems—A Case Study of London for a Warmer Climate |
| 65 | Shen | 2017 | Energy and Builidings | Impacts of climate change on US building energy use by using downscaled hourly future weather data |
| 66 | Spandagos et al. | 2017 | Applied Energy | Equivalent full-load hours for assessing climate change impact on building cooling and heating energy consumption in large Asian cities |
| 67 | Wang et al. | 2017 | Energy and Buildings | Prediction of the impacts of climate change on energy consumption for a medium-size office building with two climate models |
| 68 | Yi et al. | 2017 | Sustainable Cities and Society | Correlating cooling energy use with urban microclimate data for projecting future peak cooling energy demands: Residential neighbourhoods in Seoul |
| 69 | Chang et al. | 2018 | Building and Environment | Development of a multimedia model (IIAQ-CC) to assess climate change influences on volatile and semi-volatile organic compounds in indoor environments |
| 70 | Chen et al. | 2018 | Polish Journal of Environmental Studies | Future Climate Change on Energy Consumption of Office Buildings in Different Climate Zones of China |
| 71 | Clarke et al. | 2018 | Energy Economics | Effects of long-term climate change on global building energy expenditures |
| 72 | Croce et al. | 2018 | Climate Risk Management | The snow load in Europe and the climate change |
| 73 | Filippin et al. | 2018 | Energy and Builidings | Improvement of energy performance metrics for the retrofit of the built environment. Adaptation to climate change and mitigation of energy poverty |
| 74 | Heracleous et al. | 2018 | Energy | Assessment of overheating risk and the impact of natural ventilation in educational buildings of Southern Europe under current and future climatic conditions |
| 75 | Jeong et al. | 2018 | Sustainable Cities and Society | Projected changes to extreme wind and snow environmental loads for buildings and infrastructure across Canada |
| 76 | Jiang et al. | 2018 | Energy and Builidings | Accommodating thermal features of commercial building systems to mitigate energy consumption in Florida due to global climate change |
| 77 | Li | 2018 | Energy | Linking residential electricity consumption and outdoor climate in a tropical city |
| 78 | Lü et al. | 2018 | Building Simulation | A dynamic modelling approach for simulating climate change impact on energy and hygrothermal performances of wood buildings |
| 79 | Orr et al. | 2018 | Sci Total Environ | Wind-driven rain and future risk to built heritage in the United Kingdom: Novel metrics for characterising rain spells |
| 80 | Perez-Andreu et al. | 2018 | Energy | Impact of climate change on heating and cooling energy demand in a residential building in a Mediterranean climate |
| 81 | San Jose et al. | 2018 | Energy | Effects of climate change on the health of citizens modelling urban weather and air pollution |

**Table A1.** *Cont.*

| ID # | Author | Year | Journal | Title |
|------|--------|------|---------|-------|
| 82 | Tarroja et al. | 2018 | Applied Energy | Translating climate change and heating system electrification impacts on building energy use to future greenhouse gas emissions and electric grid capacity requirements in California |
| 83 | Triana et al. | 2019 | Energy and Builidings | Should we consider climate change for Brazilian social housing? Assessment of energy efficiency adaptation measures |
| 84 | Wang et al. | 2019 | Energy and Builidings | CESAR: A bottom-up building stock modelling tool for Switzerland to address sustainable energy transformation strategies |
| 85 | Campanico et al. | 2019 | Energy and Builidings | Impact of climate change on building cooling potential of direct ventilation and evaporative cooling: A high resolution view for the Iberian Peninsula |
| 86 | Dino et al. | 2019 | Renewable Energy | Impact of climate change on the existing residential building stock in Turkey: An analysis on energy use, greenhouse gas emissions and occupant comfort |
| 87 | Dodoo et al. | 2019 | Buildings | Effects of climate change for thermal comfort and energy performance of residential buildings in a Sub-Saharan African climate |
| 88 | Dominguez-Amarillo et al. | 2019 | Energy and Builidings | The performance of Mediterranean low-income housing in scenarios involving climate change |
| 89 | Flores-Larsen et al. | 2019 | Energy and Builidings | Impact of climate change on energy use and bioclimatic design of residential buildings in the 21st century in Argentina |
| 90 | Guarino et al. | 2019 | Energy and Builidings | Data fusion analysis applied to different climate change models: An application to the energy consumptions of a building office |
| 91 | Roshan et al. | 2019 | Building and Environment | Projecting the impact of climate change on design recommendations for residential buildings in Iran |
| 92 | Sanchez-Garcia et al. | 2019 | Energy and Builidings | Towards the quantification of energy demand and consumption through the adaptive comfort approach in mixed mode office buildings considering climate change |
| 93 | Verstraten et al. | 2019 | Building and Environment | Sensitivity of Australian roof drainage structures to design rainfall variability and climatic change |
| 94 | Zheng et al. | 2019 | Energy | Modeling the effect of climate change on building energy demand in Los Angeles county by using a GIS-based high spatial-and temporal-resolution approach |
| 95 | Andric et al. | 2020 | Energy Reports | Efficiency of green roofs and green walls as climate change mitigation measures in extremely hot and dry climate: Case study of Qatar |
| 96 | Caldas et al. | 2020 | Construction and Building Materials | Bamboo bio-concrete as an alternative for buildings climate change mitigation and adaptaton |
| 97 | Chiesa et al. | 2020 | Energy and Builidings | Contrasting climate-based approaches and building simulations for the investigation of Earth-to-air heat exchanger (EAHE) cooling sensitivity to building dimensions and future climate scenarios in North America |
| 98 | Dias et al. | 2020 | Building and Environment | The shape of days to come: Effects of climate change on low energy buildings |
| 99 | Ekolu | 2020 | Cement and Concrete Composites | Implications of global $CO_2$ emissions on natural carbonation and service lifespan of concrete infrastructures–Reliability analysis |

**Table A1.** *Cont.*

| ID # | Author | Year | Journal | Title |
|---|---|---|---|---|
| 100 | Elsharkawy et al. | 2020 | Building and Environment | The significance of occupancy profiles in determining post retrofit indoor thermal comfort, overheating risk and building energy performance |
| 101 | Figueiredo et al. | 2020 | Energy and Builidings | Country residential building stock electricity demand in future climate–Portuguese case study |
| 102 | Garshasbi et al. | 2020 | Solar Energy | Urban mitigation and building adaptation to minimize the future cooling energy needs |
| 103 | Haddad et al. | 2020 | Energy and Buildings | On the potential of building adaptation measures to counterbalance the impact of climatic change in the tropics |
| 104 | Jeong et al. | 2020 | Building and Environment | Projected changes to moisture loads for design and management of building exteriors over Canada |
| 105 | Lacasse et al. | 2020 | Buildings | Durability and Climate Change—Implications for Service Life Prediction and the Maintainability of Buildings |
| 106 | Larsen et al. | 2020 | Energy and Buildings | Climate change impacts on trends and extremes in future heating and cooling demands over Europe |
| 107 | Liu et al. | 2020 | Energy and Buildings | Effectiveness of passive design strategies in responding to future climate change for residential buildings in hot and humid Hong Kong |
| 108 | Prieto et al. | 2020 | Building and Environment | Heritage, resilience and climate change: A fuzzy logic application in timber-framed masonry buildings in Valparaíso, Chile |
| 109 | Shen et al. | 2020 | Journal of Building Engineering | An early-stage analysis of climate-adaptive designs for multi-family buildings under future climate scenario: Case studies in Rome, Italy and Stockholm, Sweden |
| 110 | Verichev et al. | 2020 | Energy and Buildings | Effects of climate change on variations in climatic zones and heating energy consumption of residential buildings in the southern Chile |
| 111 | Zhou et al. | 2020 | Building and Environment | Assessment of risk of freeze–thaw damage in internally insulated masonry in a changing climate |
| 112 | Zune et al. | 2020 | Energy and Buildings | The vulnerability of homes to overheating in Myanmar today and in the future: A heat index analysis of measured and simulated data |
| 113 | Bamdad et al. | 2021 | Energy and Buildings | Future energy-optimised buildings—Addressing the impact of climate change on buildings |
| 114 | Defo et al. | 2021 | Buildings | Effects of Climate Change on the Moisture Performance of Tallwood Building Envelope |
| 115 | Dukhan et al. | 2021 | Building and Environment | Understanding and modelling future wind-driven rain loads on building envelopes for Canada |
| 116 | Rysanek et al. | 2021 | Building and Environment | Forecasting the impact of climate change on thermal comfort using a weighted ensemble of supervised learning models |
| 117 | Vandemeulebroucke et al. | 2021 | Buildings | Factorial Study on the Impact of Climate Change on Freeze–thaw Damage, Mould Growth and Wood Decay in Solid Masonry Walls in Brussels |
| 118 | Vandemeulebroucke et al. | 2021 | Building and Environment | Canadian initial-condition climate ensemble: Hygrothermal simulation on wood-stud and retrofitted historical masonry |
| 119 | Verichev et al. | 2021 | Energy and Builidings | Adaptation and mitigation to climate change of envelope wall thermal insulation of residential buildings in a temperate oceanic climate |

**Table A1.** *Cont.*

| ID # | Author | Year | Journal | Title |
|------|--------|------|---------|-------|
| 120 | Zou et al. | 2021 | Building and Environment | A simulation-based method to predict the life cycle energy performance of residential buildings in different climate zones of China |
| 121 | Akkose et al. | 2021 | Journal of Building Engineering | Educational building retrofit under climate change and urban heat island effect |
| 122 | Alves et al. | 2021 | Energy and Builidings | The recent residential apartment buildings' thermal performance under the combined effect of the global and the local warming |
| 123 | Ascione et al. | 2021 | Energy and Builidings | Effects of global warming on energy retrofit planning of neighbourhoods under stochastic human behavior |
| 124 | Bienvenido-Huertas et al. | 2021 | Building and Environment | Analysis of climate change impact on the preservation of heritage elements in historic buildings with a deficient indoor microclimate in warm regions |
| 125 | Chen et al. | 2021 | Building and Environment | Effects of climate change on the heating indices in central heating zone of China |
| 126 | De Masi et al. | 2021 | Applied Energy | Impact of weather data and climate change projections in the refurbishment design of residential buildings in cooling dominated climate |
| 127 | Gamero-Salinas et al. | 2021 | Buildings | Passive cooling design strategies as adaptation measures for lowering the indoor overheating risk in tropical climates |
| 128 | Gaur et al. | 2021 | Building and Environment | Future projected changes in moisture index over Canada |
| 129 | Gilani et al. | 2021 | Building Research and Information | Natural ventilation usability under climate change in Canada and the United States |
| 130 | Heracleous et al. | 2021 | Journal of Building Engineering | Climate change resilience of school premises in Cyprus: An examination of retrofit approaches and their implications on thermal and energy performance |
| 131 | Rahif et al. | 2021 | Building and Environment | Simulation-based framework to evaluate resistivity of cooling strategies in buildings against overheating impact of climate change |
| 132 | Tootkaboni et al. | 2021 | Energy Reports | Analysing the future energy performance of residential buildings in the most populated Italian climatic zone: A study of climate change impacts |

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
