# Peer review of "Building Adaptation Measures Using Future Climate Scenarios—A Scoping Review of Uncertainty Treatment and Communication"

_buildings, doi:10.3390/buildings13061460_

Round 1
Reviewer 1 Report
I find the work presented to "Buildings" a good contribution to the improvement of knowledge. The topic of future climate scenarios generation and downscaling to weather files that can be used in building performance simulation is certainly very important. Please check some minor typesetting mistake or lack (e.g. line 596: "section () and ()").
As a general comment, I would like to know authors opinion (perhaps such comment can be included in "conclusion" section) about the following: If it is impossible to eliminate completely such uncertainty form future weather file (generated by any strategy), perhaps the most important thing to do is to separate uncertainties coming clearly form errors or incomplete procedures, maintaing the uncertainty range associated with possible variability in climate due to the decisions that we can make as a society. Buildings designed to respond to future climate, should be flexible and adaptable to a variety of situation we could face.
Reviewer 2 Report
The authors conduct a literature review on how climate scenarios uncertainty is addressed and communicated in studies on building adaptation. The study well highlights the limitations related to building modelling when employing climate projections, and to some extent offers concepts and indications to overcome these limitations.
However, improvements seem to be necessary both in structure and content. Specifically the result section while describing quantitatively the findings, does not give a deeper interpretation or explanation of these results. The discussion section, at the opposite, offers more insights and valid points of reflection, but this reviewer wonders to what extent these reflections are derived from the previous results. For example in section 4.3 it is not clear if table 6 and the summary of uncertainties in climate models come from the literature review. If this is the case shouldn’t these be presented in the results instead?
In detail:
- Building adaptation measures are recurrently mentioned in the text without a definition or explanation (not even in the result section where there is the possibility of giving an overview of the building measures mentioned in the reviewed articles)
- It is not clear what is intended as theoretical framework and how the sections 1 contributes to build such a framework. The confusion is even stronger when in section 2 a framework developed by Arksey and O’malley is presented.
- A more detailed description is needed on the keywords search and filtering process. Did you use all these keywords (table 2) in combination or one at a time? Can you describe better what is intended for relevance of the title/abstract. What are the criteria that define relevance?
- About the results, how did the authors select the scenarios? What is the reasoning behind and what is the general scope and scale of the modeling? For instance, is the purpose the exploration of potential climate effects or the improvement of the assessment models?
- Scenarios timeframes are mentioned in both the introduction and discussion section but not taken into consideration in the literature review. This reviewer is not convinced why the authors describe the great influence of climate scenarios in longer timeframes but then state that it is out of the scope of the review.
- Can you extend on the methods used to calculate uncertainties in the reviewed paper or other sources? This reviewer thinks that will increase the relevance of the paper.
1.1 (L 115-117) More references needed
1.3 Is unclear. Explain better what are the errors and simplifications of GCM
3.1 (L 291) What selection bias are you referring to?
4.4 (L 499-503) A few references would help
(L 591) Gender reference
(L 596) Typo (0 and 0)
Reviewer 3 Report
The topic of this review is critical for building adaptation studies, as building researchers either ignore or cannot understand the uncertainties in climate change projections. This review could bridge this gap. However, this review has two fatal weaknesses.
The first weakness is the incorrect understanding of the sources of uncertainty in climate change projections. There are three types of uncertainty: scenario uncertainty, climate response uncertainty (also referred to as model uncertainties) and natural climate variability uncertainty. The scenario uncertainty exists due to as-yet-unknown societal choices that will determine future anthropogenic emissions and land use patterns. It does not belong to any of the five levels in Table 1, as these are all model uncertainties. Refer to “1) Lee, J.-Y., et al., 2021: Chapter 4: Future Global Climate: Scenario-Based Projections and Near- Term Information. In Climate Change 2021: The Physical Science Basis. Contribution of Working Group I to the Sixth Assessment Report of the IPCC. 2) Ekström, M., et al. (2015). An appraisal of downscaling methods used in climate change research. WIREs Climate Change, 6(3), 301–319.” to gain a better understanding of the uncertainties in climate change projections.
Second, the review is expected to provide a deeper introduction to climate models, and downscaling methods so that the readers can have an idea of the sources of uncertainty. The authors should at least introduce the structure of GCMs (discretization of mathematical equations, parameterization, multi-model ensembles, etc.), RCMs, and different statistical downscaling methods (morphing, machine learning, weather generators, etc.).
Round 2
Reviewer 2 Report
The authors have substantially improved the article both in structure and content.
Reviewer 3 Report
I do not have further comments.